# Numerical Modelling of the Heat Source and the Thermal Response of an Additively Manufactured Composite during an Active Thermographic Inspection

**DOI:** 10.3390/ma17010013

**Published:** 2023-12-19

**Authors:** Arnaud Notebaert, Julien Quinten, Marc Moonens, Vedi Olmez, Camila Barros, Sebastião Simões Cunha, Anthonin Demarbaix

**Affiliations:** 1Science and Technology Research Unit, H.E.P.H Condorcet, Square Hiernaux 2, 6000 Charleroi, Belgium; arnaud.notebaert@condorcet.be (A.N.); julien.quinten@condorcet.be (J.Q.); marc.moonens@condorcet.be (M.M.); 2Research and Technological Support Department, Environmental Materials Research Association, INISMa, CRIBC, Avenue Gouverneur Cornez 4, 7000 Mons, Belgium; v.olmez@bcrc.be; 3Mechanical Engineering Institute, Federal University of Itajubá, Avenida BPS, 1303, Bairro Pinheirinho, Itajubá 37500-903, Brazil; camila.barros@condorcet.be (C.B.); sebas@unifei.edu.br (S.S.C.J.)

**Keywords:** active thermography, finite element model, radiation, composite, additive manufacturing

## Abstract

This paper deals with the numerical modelling of non-destructive testing of composite parts using active thermography. This method has emerged as a new approach for performing non-destructive testing (NDT) on continuous carbon fibre reinforced thermoplastic polymer (CCFRTP) components, particularly in view of detecting porosity or delamination. In this context, our numerical model has been developed around references containing internal defects of various shapes and sizes. The first novelty lies in the fact that the heat source used in the experimental setup is modelled exhaustively to accurately model the radiation emitted by the lamp, as well as the convection and conduction around the bulb. A second novelty concerns the modelling of the CCFRTP making up the benchmark used. Indeed, its thermal properties vary as a function of the sample temperature. Therefore, the actual thermal properties have been experimentally measured and were later used in our model. The latter then captures the material dependency on temperature. The results obtained by our model proved to be in close agreement with the experimental results on real reference points, paving the way for future use of the model to optimise experimental configurations and, in particular, the heating parameters.

## 1. Introduction

Composite materials are nowadays frequently used in a wide variety of applications, and in the aerospace industry in particular, thanks notably to their excellent strength-to-weight ratio. As a consequence, while they were initially limited to non-critical structures, composites are now in common use on primary and secondary load-carrying aircraft’s structures, such as fuselages or helicopter rotor blades. As such, one can find them on parts having large dimensions or parts showing quite complex geometries [1]. Moreover, on such critical parts, the consequences of failure could potentially be catastrophic, and all possible efforts should be enforced to avoid their potential collapse [2]. Hence the crucial need to resort to Non-Destructive Testing (NDT) in view of quality control after manufacturing or for regular inspection of these parts.

Due to their lower thermal and electrical conductivities, only a few NDT methods are applicable to composite materials [3]. In fact, with some rare exceptions, one can say that composites are currently inspected by ultrasonic testing (UT) exclusively. It is the only method certified for composite inspection in aeronautics. While UT has proven to be very reliable, it also suffers from notable drawbacks. In most cases, ultrasonic inspection requires contact with the surface to be inspected (depending on the application), and the scanning speed is slow and applied over a limited area [4]. Moreover, it can also be considered a quite polluting method, especially for large parts. Indeed, it necessitates substantial amounts of water, which is possibly contaminated after inspection if chemical agents have been added to improve the wettability of the surface. Furthermore, it is also complex to enforce on parts with complex geometries [5]. Consequently, research has been very active with the objective of finding valuable alternatives to ultrasonic inspection.

One of the emerging techniques is infrared thermography (IRT). The principle of the method resides in some form of thermal excitation of the component, which, in reaction to this excitation, emits infrared radiation, which is then captured by an infrared camera. The method can be either passive, in which case no external heat source is used (the thermal excitation being the environment itself), or active, where typically halogen lamps are used to heat up the component. In additive processes, both methods can be used to check the quality of the resulting parts. The most widely documented in the literature is the use of in-situ IRT, which is a passive method. Thermal analysis is carried out while the part is being manufactured, layer by layer. This method is widely used to optimise printing parameters or in machine learning [6,7]. Concurrently, IRT has also gained increasing interest in the context of the detection of porosity inside components made of composite materials [8]. In the latter case, one refers mainly to the active thermography method. An external heat source, such as halogen lamps or flash lamps, is used to irradiate the composite to be analysed. Thermal analysis is carried out during the transition period, when the part returns to its original temperature. The cooling effect of the air pocket causes a disturbance, enabling the defect to be localised.

Active thermography can be further subdivided into different categories depending on how the thermal excitation is brought to the component, but the most common method in the inspection of composites remains pulsed thermography, which is the heating method used in this research. As the name of the method suggests, the heat source is then turned on for a relatively short amount of time but with a high intensity, and then shut off so as to produce an impulse type of excitation. The component thus rapidly heats up when the source is on and cools down progressively after its shut-off, following the law:(1)$$T_{s}\left(t\right)-T_{s}\left(0\right)=\frac{Q}{K\rho C{\left(\pi t\right)}^{\frac{1}{2}}}$$
in which, $T_{s}$ is the surface temperature, Q is the input energy, K is the thermal conductivity of the material, ρ its density, C its specific heat, and t is the elapsed time since the end of the thermal pulse [9]. The lower thermal diffusivity of regions with defects compared to sound regions will reveal the presence of a defect, as the surface temperatures will evolve differently. However, in active thermography, one makes use of the thermal contrast (the temperature difference between a sound region and a defect region) rather than the surface temperature itself to provide accurate defect detection [1].

While proven to be very promising [10], active thermography also suffers from inherent drawbacks, such as limitations in terms of the depth of the defects that can be detected [3,10,11]. Moreover, the method must overcome two major hurdles before it can be effectively used on aircraft components. The first one resides in the dependency of the results on the heating enforcement and on the surface properties [5]. The second one concerns the difficulty of obtaining quantitative results due to the sensitivity of the results to small variations of the environment surrounding the inspected component or tiny variations of the material properties themselves.

In this context, the present research aims at developing and validating a finite element model reproducing an active thermography setup for the inspection of composite parts. A particular attention has been paid in our model to the comprehensive modeling of the heat source itself, so as to model accurately the radiation emitted by the lamp and take into account the convection and conduction around the bulb, the objective being thus to reproduce as closely as possible the actual heat source used in the experimental setup. Indeed, as highlighted in the works of Jenkins et al. [12] and Susa et al. [13] the characteristics of the halogen lamps themselves (geometry, thickness of the quartz envelope, efficiency of the reflector, etc.) should be rigorously considered to obtain numerical results that can be truly compared with experimental ones [12,13]. Next to that, particular attention has also been paid to the thermal properties of the material. Once validated, the model could potentially be used to further optimize the heating process when such an inspection is conducted on an actual part.

## 2. Finite Element Model of the Active Thermographic Inspection

In this research, the aim was to create a reliable and accurate model reproducing the entire active thermographic inspection of a benchmark component made of Continuous Carbon Fiber Reinforced Thermoplastic Polymer (CCFRTP). Several models of such inspection exist in the literature, and the authors have besides inspired their research on the previous works of Saeed et al. [4]. In this article, the innovation comes from the integral modeling of the heating source itself, because the heating process plays a crucial role in the quality of the results obtained by IRT, and from the precise characterisation of the material properties, because it has a significant influence on the way in which the simulation results agree with real experimental inspections. Our model was created using Comsol Multiphysics 6.1 software, thanks to its fluidity in combining mechanical and thermal simulations.

The benchmark used in this study was directly inspired by the works of Saaed et al. [4]. It is a rectangular plate measuring 220 mm in length, 140 mm in height, and having a thickness of 2.8 mm. It contains 16 internal defects of different sizes and shapes, the exact dimensions of which are given in Table 1, and located at different depths. These defects aim at reproducing possible delamination or porosity flaws in actual components, and the different sizes used in the benchmark yield an idea of the resolution of the active thermographic inspection. As these are internal defects, one could only resort to additive manufacturing when it came to the production of the actual benchmarks. The geometrical model of the benchmark used can be seen in Figure 1.

The composite material used for the benchmark in this simulation was realistically recreated by means of an experimental determination of its thermal properties (see Section 3.2). Experimental determination was used to obtain a value for each of the various thermal parameters of this material by obtaining a value in 5 K steps for a temperature range from 293.15 K to 333.15 K. Using these values for each step, we were able to plot a trend curve to obtain a function for calculating the specific heat and thermal conductivity of the additively manufactured composite as a function of sample temperature. These functions can then be re-used in Comsol so that the proposed numerical model can consider changes in material properties as the sample heats up or cools down during thermographic inspection. The complete material properties defined in our model are shown in Table 2.

The model developed as part of this research is a complete model that recreates the entire heat source to take direct account of most of the physical mechanisms that occur during active thermography. The geometry of the complete model is shown in Figure 2 and Table 3 shows the annotations for this Figure 2.

For the modelling of the halogen lamp, the different geometries were reproduced directly in Comsol 6.1 to avoid possible errors due to importing from other software, as errors could occur due to the number of elements and their interaction. This model of halogen lamp is made up of a tungsten filament for the heating element. This filament is surrounded by Argon, and the whole thing is placed inside a quartz wall. The aim of modelling our bulb so precisely is to accurately deduce the thermal radiation emitted by our filament, the impact that the quartz will have on this radiation, and the convection and conduction that take place inside the bulb. It is interesting to note that the quartz was modelled as a semi-transparent element to consider the real influence of its composition on the radiation (characteristics presented in Table 4). Next, the reflective part of the lamp was modelled so that the radiation emitted by the bulb could be redirected and concentrated towards the benchmark to be analysed. The last two observable elements of this heat source are the quartz disc at the lamp outlet, to take account of its impact on the radiation in a similar way to the outer envelope of the bulb, and the cylindrical aluminium part, which has an impact on thermal dispersion by channelling the radiation and heat flux. All the parameters used to model the heat source are given in Table 4.

**Table 4 materials-17-00013-t004:** Halogen source simulation characteristics.

	Bulb Characteristics	
Symbol	Simulation Parameters	Value
/	Real Light Bulb Reference	Ushio JCV240V-50WBM
P_bulb_	Power of a bulb	1000 W
H_bulb_	Total height bulb	101.6 mm
H_base_	Height of the base of the bulb	30 mm
$\emptyset_{quartz\;bulb}$	Diameter of the quartz	9.5 mm
	Reflector characteristics	
Symbol	Simulation parameters	Value
/	Real Reflector Reference	Varytec Raylight PAR 64
H_reflector_	Height of the reflector	100 mm
_∅R_ $\emptyset_{R}$	Diameter of the reflector	200 mm
/	Material of the reflector	Polished Aluminium
$\varepsilon_{R}$	Emissivity of the reflector	0.04
$\rho_{Al}$	Density of Aluminium (Reflector)	2700 kg/m^3^
$K_{R}$	Thermal Conductivity	238 W/(m × K)
$\rho_{refl}$	Specular reflexion	0.95
	Quartz Part Characteristics (semi-transparent)	
Symbol	Simulation parameters	Value
$\varepsilon_{quartz}$	Emissivity of quartz	0.1
$\rho_{d}$	Diffuse reflexion	0.08
τ	Transmission	0.82

Once the modelling was complete, a mesh with tetrahedral elements was applied. To limit the number of elements in the overall mesh and keep calculation time to a minimum, different mesh sizes were applied depending on the needs of each zone. For example, a very fine mesh was used for all the parts making up the bulb, but a normal-sized mesh was used for less sensitive parts such as the reflectors. The total number of elements for the mesh is 184,500.

## 3. Experimental Study

### 3.1. Manufacturing of the Benchmark

The experimental coupons were produced by additive manufacturing using the Anisoprint A4 printer. This printer incorporates coextrusion technology, enabling continuous carbon fibre composites to be printed. The coextrusion head mechanism can be seen in Figure 3. As regards the initial fibre state, in the coextrusion method, the reinforcing fibres are pre-saturated with thermosetting polymer. The resulting filament is then fed into a nozzle for subsequent impregnation with a thermoplastic polymer. Coextrusion tends to reduce porosity problems in the printed material, as the reinforcing fibers are continuously enveloped in the polymer [11,14].

We also chose this printer for its slicing software, which included an option for printing exactly what we wanted. This option, called ‘Masks’, allows us to determine the areas where we want to change the composition, and in our case, to include a vacuum to simulate porosity. What’s more, using this option bypasses any need to add a layer of plastic automatically added by the printer and allows us to print a part comprising only carbon fibre. The benchmark is made up of 8 plies in −45/0/45/90 orientation. The experimental benchmark can be seen in Figure 4. Additive manufacturing still lacks repeatable quality. This repeatability depends on several parameters, such as nose temperature, printing speed, ambient temperature, and build temperature. A tomography analysis of the benchmark was used to ensure that the desired air pockets were produced. This was mainly to obtain a reference caliber to validate the numerical model.

### 3.2. Material Characterisation

As it is well known, additive manufacturing can substantially modify the material properties compared to their conventional counterparts [16,17]. The material properties needed to be finely set in our numerical model so as to have the best possible match between simulation results and actual inspections.

Spoerk et al. [18] emphasize the influence of fibre orientation on both mechanical and thermal properties. Indeed, perpendicular to the fibre orientation, the thermal conductivity of the composite showed a slight increase. It is therefore important to characterize the material perpendicular to the fibres. This characterization was carried out in ply sequence, so the sequence of samples subjected to thermal characterization is −45/0/45/90 in order to consider our homogeneous composite in the numerical model.

Three samples were prepared to obtain discs with a diameter of 12.7 mm and thicknesses of 1.64, 1.80, and 1.86 mm, respectively. The surfaces of the samples are not perfectly smooth and flat due to the presence of carbon fibres in the PA matrix. It is important to take these imperfections into account when reading the results. The calculation of thermal diffusivity depends on the square of the sample thickness. Uncertainty in the value of the sample thickness due to rough or not completely flat surfaces will lead to uncertainty (error) in the determination of the thermal diffusivity and, of course, the thermal conductivity. To achieve optimum flash transfer to the samples, they are coated with a light film of Au and sputtered graphite.

The thermal conductivity of the samples was calculated directly from the thermal diffusivity measurements using a TA Instruments (New Castle, DE, USA) DLF-1600 on the samples themselves and on a well-known standard sample as a reference, in our case, Vespel. By knowing the density and the specific heat as a function of the temperature of the reference sample and the density of the samples (considered constant as a function of temperature), it is then possible to calculate the thermal conductivity by comparative measurements of thermal diffusivity between the reference and the samples. These two quantities are, in fact, linked by:(2)$$K\left(T\right)=\alpha \left(T\right)\times \rho \left(T\right)\times Cp\left(T\right),$$
where K(T) is the thermal conductivity in W/(m × K), α(T) is the thermal diffusivity in m^2^/s, ρ(T) is the density in kg/m^3^ and Cp(T) is the specific heat capacity in J/(kg × K). Finally, the density of the sample was measured by Archimedes’ method on a larger printed piece.

### 3.3. Thermographic Inspection on Additively Manufactured Benchmarks

In order to validate the model created and presented in Section 2, we carried out an experimental test to compare the results with those obtained by numerical simulation. For this experimental part, the set-up used (Figure 5) consisted of two 1000 W halogen lamps, each installed on a tripod, with the bulbs located at a distance of 1 m from the benchmark to be analysed. The specimens were then heated up with a 10 s pulse of the halogen lamps, and the thermal response in reflexion was then recorded with a FLIR T865 infrared camera, a focus lens of 10 mm (42°), and a NETD < 0.03 °C (<30 mK) thermal resolution. The thermal camera operates in a spectral band ranging from 7.5 μm to 14 μm and is located at the same distance as the lamps and at the same height as the benchmark under study. The software used to analyse the thermal response was FLIR Tools 6.4. The duration of observation of the thermal response was 45 s after heating, i.e., a total recording of 55 s given the duration of the pulse.

For the experimental part, we were able to plot the first temperature curves for the two evaluation zones directly in the FLIR Tools programme (the programme used to read the thermograms created with the thermal camera). From these curves, we were able to export all the data used by the program to plot them. It should be noted that we obtained a sufficient number of data points with a temperature for each evaluation zone in steps of 0.1 s. From this data, we were able to calculate the thermal contrast, plot our curve, and determine the maximum of this curve and the instant at which it appears with precision.

## 4. Results

The model developed in this research was used to reproduce the thermal evolution of the benchmark when heated by halogen lamps (as described in Section 2 and Section 3) during an active thermography inspection. A time step of 1 s was defined so as to obtain a total of 55 thermograms, yielding a clear view of the temperature evolution during the heating up and cooling down of the benchmark. One of the thermograms, here at the end of the 10-s heating pulse, is shown in Figure 6.

On this thermogram, we can observe the inhomogeneity of the surface temperature of the coupon, hence already revealing some of the defects. In practice, however, defect detection is actually based on thermal contrast, which compares the temperature of an area containing a defect with the temperature of an adjacent healthy area. Thermal contrast is thus obtained using Equation (3):(3)$$T_{c}\left(t\right)=T_{d}\left(t\right)-T_{s}(t)$$
where T_c_(t) is the thermal contrast, T_d_(t) is the temperature of the zone with the defect, and T_s_(t) is the temperature of the healthy area at time t [4].

In view of comparison with experimental data, the thermal contrasts of two defects among the sixteen were analysed. The considered defects are shown in Figure 5. In the numerical model, the two lamps are exactly focused on the centre of the plate. In the experimental setup, however, it was noticed that there was a slight offset in the focus of the lamps. This induced a thermal inhomogeneity in the experimental response, which has nothing to do with the thermal behaviour of the specimen. To avoid this inhomogeneity, post-processing algorithms are used to obtain a homogeneous result. Several filters can be used on a thermogram, resulting in slight differences between results. As the numerical model is to be used to optimize heating parameters, thermal analysis is applied to the raw thermograms. The two selected defects are the ones where this thermal inhomogeneity is felt the least, hence enabling a proper comparison between experimental data and numerical results.

Regarding the squared-shaped defect, the surface temperature is recorded at its centre (located at a height of 37.5 mm from the base of the plate—bottom line on Figure 7), but also at 10 mm on the right of the defect. The latter then serves as the healthy adjacent zone for the computation of the thermal contrast. The same strategy is used for the circular-shaped defect (see Figure 7). One should remember that the two defects used to analyse the results are not located at the same depth. In fact, the square-shaped defect is located deeper than the round defect (a difference of 0.25 mm in depth). With these definitions of the evaluation zones in mind, one will then be able to obtain the thermal contrast for both defects and follow their evolution with time. As exactly the same evaluation zones were used in the experimental part (see Section 3.3), one will thus be able to compare the results of our model with the ones obtained experimentally. In the analysis of our results and the comparison between the experimental and the simulation, errors were calculated with regard to the maximum contrast peaks and the time at which these peaks occurred. To calculate these errors, the following formula was used:(4)$$Error=\frac{T_{exp}-T_{sim}}{T_{exp}}$$
where T_sim_ is the temperature taken from the simulation results and T_exp_ is the temperature taken from the experimental results.

The evolution of thermal contrasts, both experimental and numerical, for the round-shaped defect is shown in Figure 8. It is notable that the evolutions are quite similar. Both curves present a peak thermal contrast, which is respectively 2.9 K and 3.2 K for the numerical and experimental results. The numerical model hence yields an error of 9.3% in terms of maximum contrast. One should also note that the peaks do not occur exactly at the same time (16 s, compared to 17.5 s in the experimental case). This difference in the time at which the peak contrast appears is due to the transient effect present in the experimental model. In fact, experimentally, we have a transient effect to reach 1000 W of heating, a transient effect that is not included in the experimental model, which means that in the simulation, maximum power is reached more quickly (instantaneously), and therefore peak contrast is reached slightly earlier than in the experimental model. Moreover, at t = 2 s and less clearly at t = 12 s, changes in the curvature of the numerical curve can be observed. These time steps directly correspond to the switching on and off of the heating sources. The modelling of the physics of the lamps and the behaviour of the material is therefore sufficiently accurate to capture the effects of the start and end of the heat pulse. The reason why this phenomenon is only observed on the curve from the simulation and not on the experimental curve is also due to the transient effect present only in the experimental curve. In the simulation, from the start of the pulse, the plate receives a direct flow of 1000 W, which is not the case in the experiment, where the power arrives more gradually and is not instantaneous. The same phenomenon occurs at the end of the pulse. In this case, in simulation, the power drops instantaneously from 1000 W to 0 W, which is not exactly the case in experiments because of the transient effect. Finally, starting from t = 22 s until the end, it can be clearly seen that the thermal contrast in our simulation decreases more slowly than the experimental thermal contrast. This is mainly explained by the offset in focus of the experimental lamps, while, as mentioned above, the focus is exactly at the center of the plate in the numerical model. While the error might appear substantial, as the two curves seem significantly apart from each other at the end, the difference in thermal contrast is only a few tenths of a degree (0.28 K). In terms of absolute temperatures computed by the model, this corresponds to a tiny error.

Now let us look at the graph for the square fault (Figure 9). The graph also shows a similar pattern, but with a more noticeable difference at the peak of the thermal contrast. This greater difference between the experimental curve and the digital curve is due to the place where the radiation was concentrated experimentally. Indeed, as explained above, in the experiment, the beam was more concentrated and did not strike exactly at the centre of the plate but slightly to the right. Given that in this case we’re on a defect closer to the real impact zone in the experiment, we’re also in a zone where the influence of this slight difference between the simulation and the experiment is felt more strongly, which gives us a greater thermal contrast. As a result of this influence, we have a greater difference than previously for the maximum contrast value, with this time an error of 17% between the simulation and the experiment, the origin of which comes from the same cause as before. With regard to the time at which the peak occurs, this error remains in the same order of magnitude, since in the simulation the peak appears at the 18th s and after around 19 s in the experiment, giving an error of 5%. As before, we observe the same two movements in the numerical contrast curve for the same reasons as in the previous case, but we also observe the impact of the start of the pulse on the experimental curve where we have a negative thermal contrast, which translates into a higher temperature in the healthy zone at the start of the pulse. The fact that this contrast is negative in the experimental curve is also a consequence of the impact zone of the light beam in the experimental curve compared with the simulation curve, which explains the difference between the two curves. Finally, after approximately 23 s, we observe the same phenomenon at the end of the evolution as in the previous case, which is undoubtedly due to the same causes.

## 5. Conclusions

In this article, we have developed a numerical model of the entire active thermography process. This model simulates active thermography by reproducing all the elements making up the experimental device, which means creating a simulation that directly integrates a model of the heat source used, which in our case are halogen lamps. The aim of this more realistic simulation is to take direct account of all the major physical phenomena that occur during thermography. A numerical model of active thermography is a tool with several advantages, because once created, such a model makes it possible to quickly and economically determine the optimum parameters for carrying out active thermography on thermoplastic composites with continuous carbon fibres to obtain improved sensitivity, increased detection of defects, or improved sensitivity to defects. With the results obtained, we can see that our model remains faithful to reality, obtaining results that are generally close to those obtained experimentally. The slight differences can be explained by the experimental setup, which makes it difficult to ensure the exact position of the halogen lamps, while in the numerical model, the position of the lamps is exactly determined. It would therefore be necessary to improve our experimental method to guarantee closer agreement between the experimental parameters and the numerical parameters.

## Figures and Tables

**Figure 1 materials-17-00013-f001:**
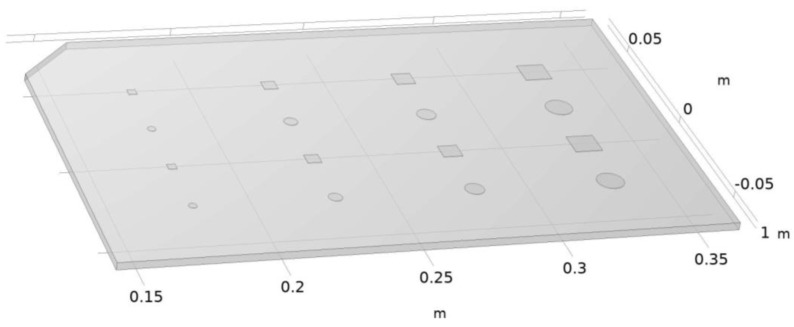
Benchmark geometry.

**Figure 2 materials-17-00013-f002:**
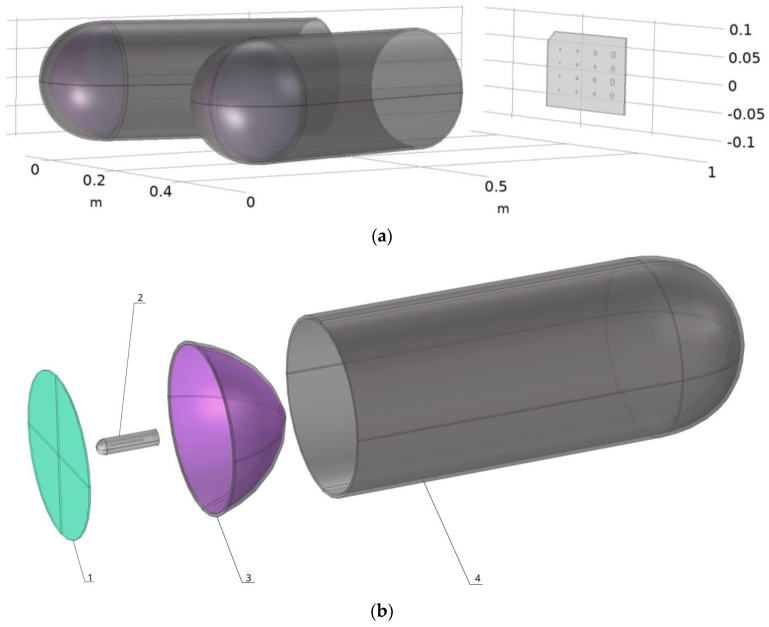
Modelling the different geometries that make up the numerical simulation (**a**) complete set-up modelling (**b**) exploded view of the lamp.

**Figure 3 materials-17-00013-f003:**
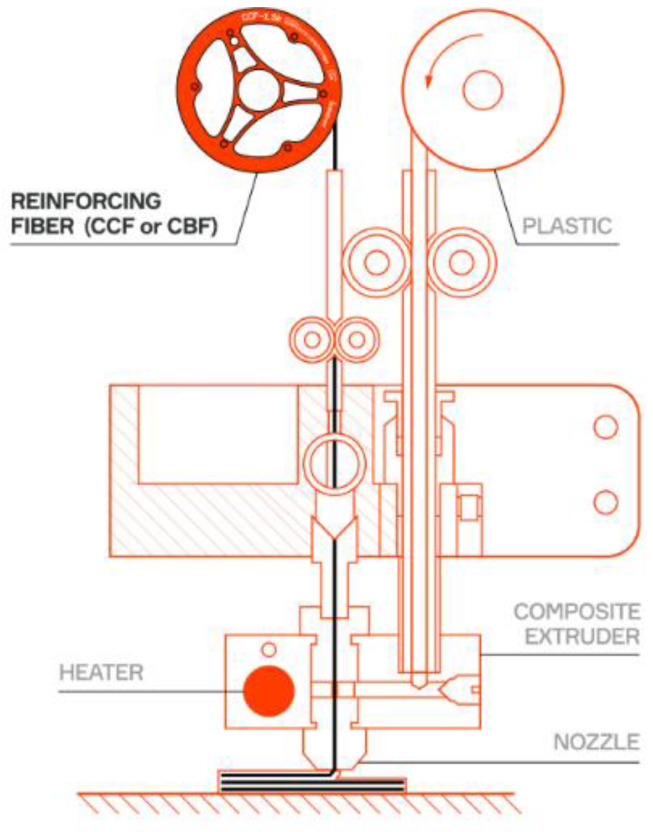
Coextrusion print head (authorized by Anisoprin [15]).

**Figure 4 materials-17-00013-f004:**
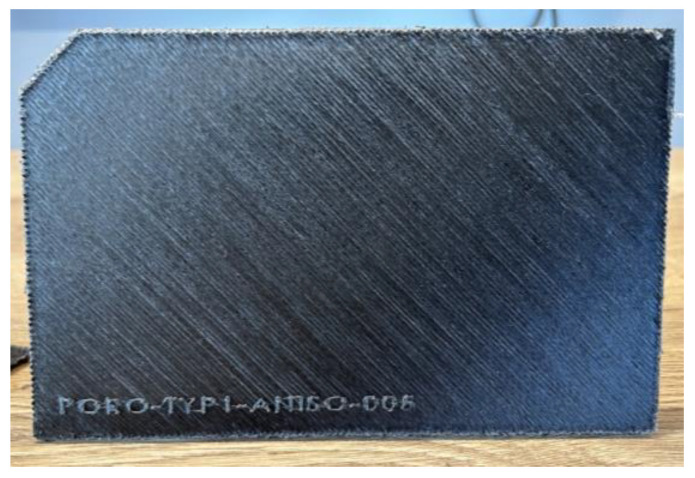
Experimental benchmark.

**Figure 5 materials-17-00013-f005:**
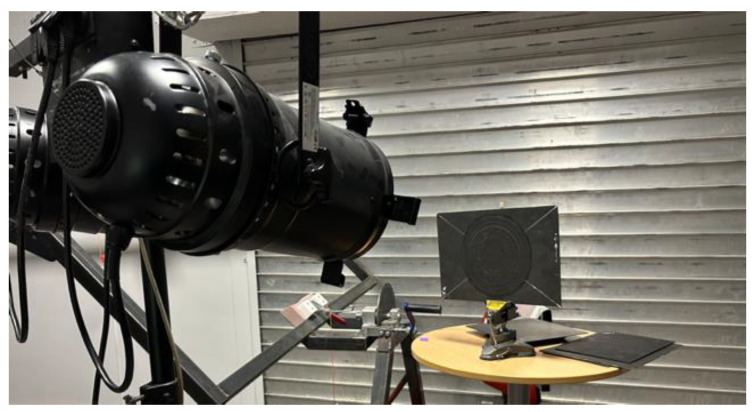
Experimental set-up.

**Figure 6 materials-17-00013-f006:**
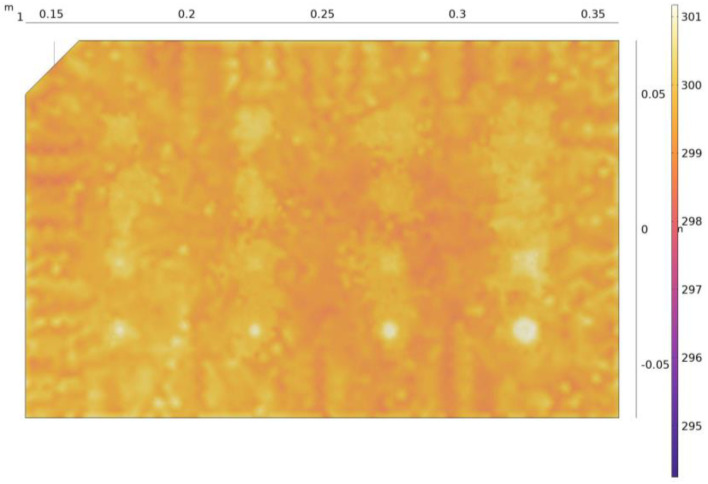
Thermogram (as computed by our model) at the end of the heat pulse.

**Figure 7 materials-17-00013-f007:**
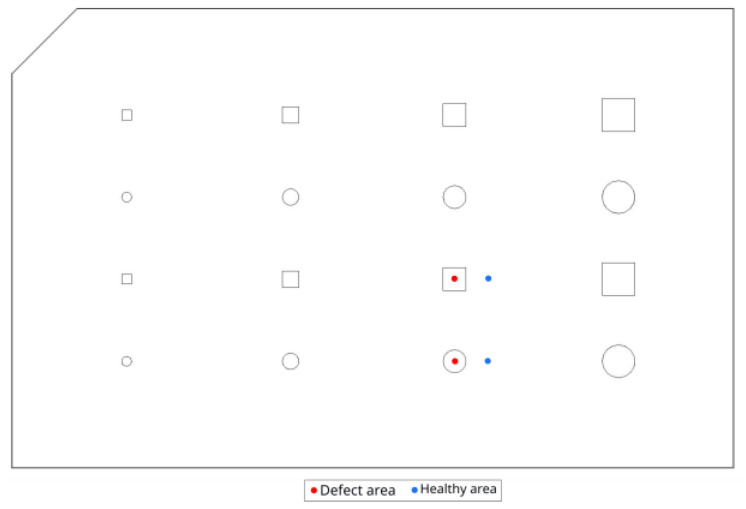
Location of evaluation zones.

**Figure 8 materials-17-00013-f008:**
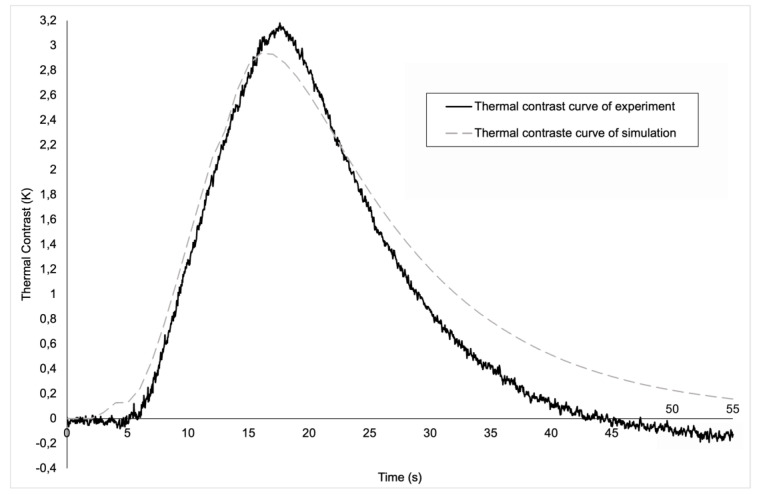
A graphic of the evolution of the thermal contrast for the round defect.

**Figure 9 materials-17-00013-f009:**
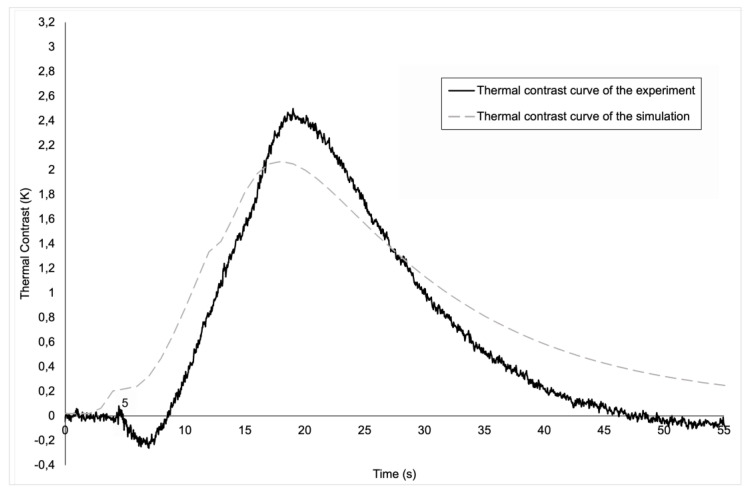
A graphic of the evolution of the thermal contrast for the square defect.

**Table 1 materials-17-00013-t001:** Defect characteristics.

Row	Depth	Shape	Defect Size
1	2 mm	Square	Side length: 3 mm, 5 mm, 7 mm, 10 mm
2	1.75 mm	Circular	Diameter: 3 mm, 5 mm, 7 mm, 10 mm
3	1.5 mm	Square	Side length: 3 mm, 5 mm, 7 mm, 10 mm
4	1.25 mm	Circular	Diameter: 3 mm, 5 mm, 7 mm, 10 mm

**Table 2 materials-17-00013-t002:** Material characteristics.

	Benchmark Characteristics	
Symbol	Simulation Parameters	Value
/	Material	CCF 1.5 K and CFC PA
$\varepsilon_{CCFRTP}$	Emissivity	0.98
$\rho_{CCFRTP}$	Density	989 kg/m^3^
C_p,CCFRTP_	Specific Heat (CCFRTP)	5.6714 × T − 234.84 J/(kg × K)
$K_{CCFRTP}$	Thermal Conductivity	0.001 × T + 0.0249 W/(m × K)

**Table 3 materials-17-00013-t003:** Material characteristics.

Description of the Lamp Part	
Number	Description
1	Quartz plate at the lamp outlet
2	Bulb with filament
3	Reflector
4	Aluminium frame (painted black)

## Data Availability

Data are contained within the article.

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
