# Peer review of "Numerical Modelling of the Heat Source and the Thermal Response of an Additively Manufactured Composite during an Active Thermographic Inspection"

_materials, 2023, doi:10.3390/ma17010013_

Round 1

Reviewer 1 Report

Comments and Suggestions for Authors

Dear authors,

Thanks for your interest to submitting your paper to Materials Journal. We appreciate your dedication. However, this paper proposed a numerical method for thermographic inspection, but all methodologies are conducted through commercial software and the detailed algorithm/theory is not adequately provided and discussed. Didn't see the novelty or the scientific soundness in this paper. The conclusions are not supported by the results. Accordingly, after careful consideration, I don't recommend the journal "Materials" to accept this paper. The detailed reasons are shown below.

1.    The experimental setup of thermographic inspection was not described clearly by the authors. No figure to describe the test setup and it’s very hard for readers to compare the test setup and simulation setup side by side. Accordingly, the readers can’t make sure the robustness and accuracy of the test setup and the simulation setup are good enough respectively and if both are at the same level or not.

2.    Figure 2 needs more annotations to explain the set up.

3.    How the temperature contrast data read/measured from the real tests is not clear or provided by the authors, which is significant to show the scientific soundness of the test results.

4.    The authors provide a sample with 4x4 defect dimensions, but only pick up 2 of them arbitrarily to show the results. Also, only one printed sample is not robust enough. Again, statistic data should be provided, instead of arbitrary data points from one sample only.

5.    Generally, the content is very difficult to understand, especially in the Results section.

6.    Reference format is not consistent through the whole paper and there is empty reference.

 Thanks!

Comments on the Quality of English Language

Very hard to understand the description. Extensive editing of English language required.

Author Response

The responses to the comments from all the reviewers can be consulted in the attached document in PDF format. The changes made are visible in the new version of the manuscript, with the relevant parts highlighted in blue font. 

Reviewer 2 Report

Comments and Suggestions for Authors

My comments are attached in the word document.

Comments on the Quality of English Language

Extensive editing required. 

Author Response

(The authors gave the same response as above.)

Reviewer 3 Report

Comments and Suggestions for Authors

This paper developed a numerical model of the entire active thermography process. This model simulates active thermography by reproducing all the elements making up the experimental device.

Some comments:

1. How do you calculate the error in figure 6 and 7?

2. There is big error at the tail of both figure 6 and 7. Is there any reason for it?

The accuracy need to be improved in my opinion.

Author Response

(The authors gave the same response as above.)

Reviewer 4 Report

Comments and Suggestions for Authors

The authors present a numerical modelling strategy for the non-destructive testing of composite parts by active thermography. The work is novel and the paper is well written. However, the paper can be improved by addressing minor comments. 

1. Authors should elaborate on how the constitutent properties of carbon fibers will affect the macroscopic mechanical response and describe the methods to obtain the coefficients for numerical modeling. 

2. The affect of additive manufacturing patterns on the overall thermal response could be briefly discussed. 

3. Relevant papers on thermal analysis of additively manufactured papers should be cited. 

Pokkalla, D.K., Hassen, A.A., Heineman, J., Snape, T., Arimond, J., Kunc, V. and Kim, S., 2022, October. Thermal Analysis and Design of Self-Heating Molds Using Large-Scale Additive Manufacturing for Out-of-Autoclave Applications. In ASME International Mechanical Engineering Congress and Exposition (Vol. 86649, p. V02BT02A007). American Society of Mechanical Engineers.

Jo, E., Liu, L., Ju, F., Hoskins, D., Pokkalla, D., Kunc, V., Vaidya, U. and Kim, P., 2022. The design of layer time optimization in large scale additive manufacturing with fiber reinforced polymer composites. Oak Ridge National Lab.(ORNL), Oak Ridge, TN (United States).

Author Response

(The authors gave the same response as above.)

Round 2

Reviewer 1 Report

Comments and Suggestions for Authors

line number is mixed with table. Such as lines 141 to 143 and lines 175 to 182.

Comments on the Quality of English Language

Needs to be improved.

Reviewer 2 Report

Comments and Suggestions for Authors

The revised manuscript may be accepted in its current form.

Comments on the Quality of English Language

Language and style are OK.